# PQP: A one-shot collaborative method for post-quantization pruning of LLMs

## Abstract

Achieving large-scale model compression at a lower cost without significantly compromising performance remains an important challenge in the deployment of LLMs. To address this, we propose Post-Quantization Pruning (PQP), a framework that enables effective synergy between quantization and pruning. Specifically, we introduce a pruning metric tailored for post-quantization scenarios. In addition, we integrate the statistical properties of quantization error into the loss function for block-wise pruning, allowing the model to simultaneously eliminate weights that are unimportant or heavily affected by quantization noise. The PQP method integrates post-training quantization and one-shot pruning, reducing the compressed model's storage requirements to just 1/8 of the original size, with minimal impact on performance. The entire compression process is lightweight, easy to implement, and does not require fine-tuning. Experimental results on LLaMA and LLaMA-2 models demonstrate that PQP consistently outperforms existing methods in compressing models of various scales. Our code is available at `https://anonymous.4open.science/r/PQP`.

## 1 Introduction

Large Language Models (LLMs) (Touvron et al., 2023a;b; Zhang et al., 2022; Zeng et al., 2023) have had a revolutionary impact on natural language processing tasks, with a wide range of applications. However, as the scale of LLMs continues to grow (*e.g.*, parameter counts increasing from billions to trillions), their demands for computation, storage, and bandwidth rise dramatically, which affects their feasibility for deployment. These challenges have driven researchers to develop faster and more efficient compression techniques for LLMs(Tang et al., 2024; Zeng et al., 2025).

In recent years, numerous model compression algorithms have been proposed. Among them, quantization and pruning have become representative techniques due to their effectiveness and simplicity. Model quantization (Liu et al., 2025a) involves converting high-precision floating-point weights in neural networks into lower-precision weights. Depending on whether training is involved, quantization methods are typically classified into Quantization-Aware Training (QAT) and Post-Training Quantization (PTQ). QAT approaches (Liu et al., 2023; Guan et al., 2024) can maintain high model accuracy even under low-bit quantization, but they require considerable computational resources. PTQ methods (Frantar et al., 2023; Lee et al., 2024; Edalati et al., 2025) are more efficient as they avoid retraining, but often suffer notable performance drops as bit width decreases.

Model pruning (Zhang et al., 2024a; Huang et al., 2025) reduces model size by identifying and removing redundant parameters. Recently, one-shot pruning methods (Frantar & Alistarh, 2023; Zhang et al., 2024b; Liu et al., 2025b) have been proposed for LLM compression, removing the need for retraining or fine-tuning and thus greatly improving efficiency.

Although one-shot quantization and pruning are effective individually, neither can achieve very high compression ratios without incurring significant performance degradation or resource overhead. Achieving higher compression rates with lower resource consumption and minimal performance loss remains an open challenge in model compression research.

A straightforward idea is to combine quantization and pruning. Many existing studies on quantization and pruning claim that their methods can be directly integrated with other types of compression techniques. However, can such a straightforward combination truly deliver high compression ratios

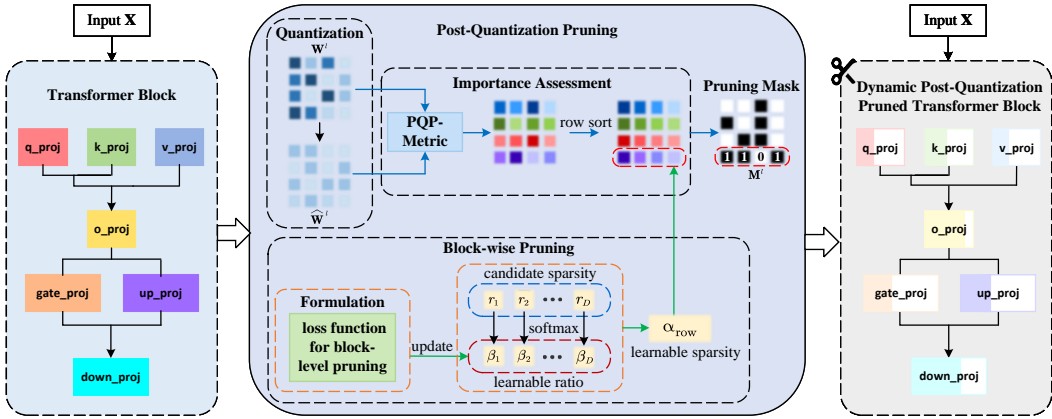

Figure 1: Workflow of the Proposed PQP Method.

while maintaining strong performance and low cost? (Frantar & Alistarh, 2022; 2023) explored this direction by directly combining quantization and pruning. (Guo et al., 2024) performed 8-bit quantization on pruned weights by addressing outliers, but still observed accuracy degradation under low-bit quantization. Although they improved compression ratios, the models suffered from notable performance drops and higher computational costs. We argue that such degradation is due to the compounded effects of errors introduced by different compression strategies. Therefore, mitigating the impact of these errors and achieving an effective, principled integration of quantization and pruning remain worthwhile pursuits.

To address this, we propose a novel Post-Quantization Pruning (PQP) method that seamlessly combines PTQ with one-shot pruning, preserving the key advantage of both: no retraining or fine-tuning. To mitigate the impact of quantization error, we design a tailored pruning metric for post-quantization pruning and incorporate the statistical characteristics of quantization error into the pruning loss function. Our PQP approach achieves higher compression with minimal performance degradation and resource cost.

Overall, this work makes the following contributions. First, we propose PQP, a practical and efficient framework that synergistically combines quantization and pruning. PQP enables one-shot model compression with a high compression ratio, low performance degradation, and minimal resource overhead. Second, we introduce a new pruning metric specifically designed for quantized models. By leveraging weight information from both the original and quantized networks, this metric effectively reduces the performance loss typically caused by pruning. Third, we explicitly model quantization error and incorporate its statistical properties into the pruning objective, further mitigating degradation during compression. Finally, extensive experiments on LLaMA and LLaMA-2 demonstrate the effectiveness of PQP, showing consistent and significant improvements over existing methods.

## 2 RELATED WORK

### 2.1 POST-TRAINING QUANTIZATION (PTQ)

PTQ methods have gained increasing attention for compressing LLMs, as they eliminate the need for retraining and significantly reduce computational costs compared to QAT. AdaRound (Nagel et al., 2020) and AdaQuant (Hubara et al., 2021) are representative PTQ approaches. Both methods employ lightweight post-training quantization, leveraging a calibration dataset to jointly adjust layer-wise weights and activations, thereby minimizing quantization error. OBQ (Frantar & Alistarh, 2022) mitigates quantization-induced accuracy loss by performing layer-wise error compensation. GPTQ (Frantar et al., 2023) builds on OBQ and improves efficiency by introducing a faster method for computing the Hessian inverse. For the emergent outliers, LLM.int8() (Dettmers et al., 2022) introduces mixed-precision quantization for LLMs by isolating and quantizing outlier features with higher bit-width. SqueezeLLM (Kim et al., 2024) further mitigates performance degradation by storing sensitive weights and outliers in a sparse format. OAC (Edalati et al., 2025) builds on this

Table 1: One-shot pruning evaluation metrics for LLMs, where $\mathbf{G}$ denotes gradient information, $mms(\cdot)$ refers to min-max scaling, and PM stands for Pruning Metric.

| Method | W | G | PM |
|---|---|---|---|
| SparseGPT(Frantar & Alistarh, 2023) | ✓ | ✗ | $[\lvert\widehat{\mathbf{W}}\rvert^2/diag(\mathbf{H}^{-1})]_{ij}$ |
| Wanda(Sun et al., 2024) | ✗ | ✗ | $\lvert\widehat{\mathbf{W}}\rvert \cdot \lVert\mathbf{X}_j\rVert_2$ |
| BESA(Xu et al., 2024) | ✗ | ✗ | $\lvert\widehat{\mathbf{W}}\rvert \cdot \lVert\mathbf{X}_j\rVert_2$ |
| RIA(Zhang et al., 2024b) | ✗ | ✗ | $(\lvert\widehat{\mathbf{W}}_{ij}\rvert/\sum\lvert\widehat{\mathbf{W}}_{*j}\rvert + \lvert\widehat{\mathbf{W}}_{ij}\rvert/\sum\lvert\widehat{\mathbf{W}}_{i*}\rvert) \cdot \sqrt{\lVert\mathbf{X}_j\rVert_2}$ |
| Pruner-Zero(Dong et al., 2024) | ✗ | ✓ | $\lvert\widehat{\mathbf{W}}\rvert \cdot \lvert\widehat{\mathbf{W}}\rvert \cdot mms(\lvert\mathbf{G}\rvert)$ |
| BaWA(Liu et al., 2025b) | ✗ | ✗ | $(\lvert\widehat{\mathbf{W}}_{ij}\rvert \cdot (\lVert\widehat{\mathbf{W}}_j\rVert_2^{\theta_1})^{-1} + (\lVert\widehat{\mathbf{W}}_i\rVert_2^{\theta_2})^{-1} \cdot \lvert\widehat{\mathbf{W}}_{ij}\rvert) \cdot \lVert\mathbf{X}_j\rVert_2^{\theta_3}$ |
| PQP(ours) | ✗ | ✗ | $\lvert\widehat{\mathbf{W}}\rvert \cdot mms\left(\text{PM}\left(\mathbf{W}\right)\right)$ |

by overcoming SqueezeLLM's limitation in handling extremely low-bit precision. However, PTQ methods still struggle to achieve high compression ratios without significant performance loss.

## 2.2 ONE-SHOT PRUNING

One-shot pruning methods are widely used for their ability to remove model parameters without requiring retraining or fine-tuning. The effectiveness of these methods largely depends on the design of importance metrics that guide pruning decisions. SparseGPT (Frantar & Alistarh, 2023) is the first to introduce a one-shot pruning approach specifically for LLMs, utilizing second-order information from the Taylor expansion to compensate for pruning-induced errors without any training. Wanda (Sun et al., 2024) simplifies SparseGPT by approximating the inverse Hessian with a diagonal matrix, significantly reducing the computational cost. RIA (Zhang et al., 2024b) reconsiders the role of weight magnitude from the perspective of weight importance and proposes a refined pruning criterion. GBLM-Pruner (Das et al., 2023) and Pruner-zero (Dong et al., 2024) enhance pruning metrics by incorporating first-order information (i.e., gradients) derived from the Taylor expansion. DSnoT (Zhang et al., 2024c) introduces a grow-prune strategy to dynamically reconstruct pruning masks, while BESA (Xu et al., 2024) implements block-wise dynamic sparsity allocation through learnable sparsity levels. BaWA (Liu et al., 2025b) further introduces a one-shot pruning metric that balances weight and activation distributions, enabling more effective and efficient pruning.

However, these methods all rely on the model's original weights. As shown in Table 2, directly applying such importance metrics after quantization yields suboptimal results, underscoring the need for metrics specifically designed for quantized weights.

## 3 METHODS

In this section, we introduce a novel PQP method that enables higher compression ratios with minimal computational overhead and without significantly compromising model performance. The overall PQP workflow is illustrated in Figure 1. The left side of the figure shows a typical Transformer block from the original model, while the right side presents the block after applying PQP. The middle section illustrates the proposed quantization-pruning pipeline. We begin by quantizing the trainable parameters of the LLM. Next, we assess parameter importance and propose a dedicated importance metric specifically designed for pruning in post-quantization settings. We then formulate a loss function for block-wise pruning and simplify it by leveraging the statistical characteristics of quantization errors. Finally, we incorporate this loss function with a learnable sparsity (pruning rate) to effectively guide the model pruning process.

## 3.1 PRELIMINARY

**Layer-wise post-quantization pruning:** Given the layer input $\mathbf{X}^l \in \mathbb{R}^{L_s \times C_{in}}$, the weight matrix $\mathbf{W}^l \in \mathbb{R}^{C_{in} \times C_{out}}$, and its corresponding quantized weights $\widehat{\mathbf{W}}^l \in \mathbb{R}^{C_{in} \times C_{out}}$ in the $l$-th layer of an LLM, where $L_s$, $C_{in}$ and $C_{out}$ are input sequence length, weight input dimension, and weight output dimension respectively, the objective function for layer-wise post-quantization pruning, applicable to both self-attention and feed-forward modules, can be expressed as:

$$\underset{\mathbf{M}^l}{\operatorname{argmin}} \left\lVert \mathbf{X}^l\mathbf{W}^l - \mathbf{X}^l(\widehat{\mathbf{W}}^l \odot \mathbf{M}^l) \right\rVert_F^2, \quad s.t. \quad 1 - \frac{1}{N}\lVert\mathbf{M}^l\rVert_0 = p. \tag{1}$$

where $p$ represents the target pruning ratio, $N$ denotes the total number of elements in the $l$-th layer, and $\|\cdot\|_0$ is the $\ell_0$-norm, which returns the total count of non-zero elements. $\mathbf{M}^l \in \mathbb{R}^{C_{in}\times C_{out}}$ denotes the binary pruning mask, where an entry of 0 indicates that the corresponding element in $\widehat{\mathbf{W}}^l$ is pruned, and an entry of 1 indicates that it is retained. It is worth noting that the layer-wise post-quantization pruning method in equation 1 is tailored for quantized models, distinguishing it from conventional one-shot pruning approaches.

## 3.2 IMPORTANCE ASSESSMENT

**Motivation:** In Table 1, we summarize existing one-shot pruning metrics, most of which claim to be directly compatible with other model compression strategies such as quantization. However, when these methods are applied to assess the importance of weights in quantized models, they often overlook the impact of quantization errors. We argue that such errors can undermine the reliability of pruning importance scores, particularly in low-bit quantization scenarios, where substantial quantization errors may significantly distort the ranking of weight importance.

**PQP Metric:** We propose a novel pruning metric specifically designed for quantized weights, aiming to mitigate the impact of quantization errors and effectively assess their importance. The design of this metric is inspired by that in Pruner-Zero (Dong et al., 2024). Pruner-Zero regularizes the gradient $\mathbf{G}$ and uses the product of $\mathbf{G}$ and $\mathbf{W}$ as the pruning metric, where the regularized gradient acts as a scaling factor that reflects the importance of $\mathbf{W}$. Inspired by this idea, we seek a similar scaling factor for the quantized weights $\widehat{\mathbf{W}}$ to offset quantization errors. To do this, we regularize the original weights and use the result as a scaling factor. This approach introduces quantization information while avoiding the high computational cost of gradient calculations in Pruner-Zero. Accordingly, we propose a new pruning metric for quantized weights, defined as:

$$PQP\text{-}Metric = |\widehat{\mathbf{W}}| \times mms(PM(\mathbf{W})), \tag{2}$$

where $|\widehat{\mathbf{W}}|$ denotes the element-wise absolute value of the quantized weights, and $mms(\cdot)$ refers to the regularization obtained by applying Min-Max Scaling to $PM(\mathbf{W})$, which scales the data to the range [0,1]. $PM(\mathbf{W})$ represents the result of applying the Pruning Metric (PM) from Table 1 to the original weight matrix $\mathbf{W}$. (Based on Table 5, we have selected RIA as the pruning metric for practical use.) Obviously, this metric leverages both the original weights' importance and the quantized weights' magnitude to offer a comprehensive assessment of parameter importance in the quantized model.

## 3.3 FORMULATION OF POST-QUANTIZATION PRUNING

Quantization process introduces significant quantization errors, which undoubtedly negatively impact post-quantization pruning. Thus, minimizing the effect of quantization error is a key challenge that post-quantization pruning must overcome. To address this, we parametrize the quantization error and incorporate it into the objective function. By statistically modeling the quantization error and leveraging its statistical properties, we can further simplify the objective function.

Specifically, we first parameterize the quantization error and integrate it into the objective function equation 1, yielding:

$$\underset{\mathbf{M}^l}{\arg\min} \left\| \mathbf{X}^l\mathbf{W}^l - \mathbf{X}^l((\widehat{\mathbf{W}}^l - \mathbf{B}^l)\odot \mathbf{M}^l) \right\|_F^2, \tag{3}$$

where $\mathbf{B}^l := \widehat{\mathbf{W}}^l - \mathbf{W}^l$ represents the layer-wise quantization error, which quantifies the difference between $\widehat{\mathbf{W}}^l$ and $\mathbf{W}^l$. To analyze the impact of quantization errors on model performance, we define the objective in equation 3 as $J$, which can be rewritten as:

$$J = \underbrace{\left\| \mathbf{X}^l\mathbf{W}^l - \mathbf{X}^l(\widehat{\mathbf{W}}^l \odot \mathbf{M}^l) \right\|_F^2}_{J_1} + \underbrace{\left\| \mathbf{X}^l(\mathbf{B}^l \odot \mathbf{M}^l) \right\|_F^2}_{J_2} + \underbrace{2\text{Tr}((\mathbf{X}^l\mathbf{W}^l - \mathbf{X}^l(\widehat{\mathbf{W}}^l \odot \mathbf{M}^l))^\top \mathbf{X}^l(\mathbf{B}^l \odot \mathbf{M}^l))}_{J_3}.$$

$$\tag{4}$$

Here, the quantization error $\mathbf{B}^l$ can be viewed as the error introduced by rounding high-precision data during the quantization process. Therefore, we can model it statistically as a zero-mean Gaussian

distribution with variance $\sigma_b^{l\,2}$: $\mathbf{B}^l \sim N(0, \sigma_b^{l\,2})$. Based on the statistical properties of the Gaussian distribution for $\forall B_{rj}^l \in \mathbf{B}^l$, we have $\mathbb{E}[B_{rj}^l] = 0$.

The pruning objective function in equation 4 is relatively complex and may lead to difficulties in model convergence. To simplify the objective, we leverage the statistical properties of the quantization error $\mathbf{B}^l$. In particular, we redefine the objective function as

$$J := J_1 + J_2 + \mathbb{E}[J_3]. \tag{5}$$

For ease of derivation, $J_3$ can be rewritten as: $2\mathrm{Tr}(\mathbf{A}^\top \mathbf{C})$, where $\mathbf{A} = \mathbf{X}^l \mathbf{W}^l - \mathbf{X}^l(\widehat{\mathbf{W}}^l \odot \mathbf{M}^l) \in \mathbb{R}^{L_s \times C_{out}}$, and $\mathbf{C} = \mathbf{X}^l(\mathbf{B}^l \odot \mathbf{M}^l) \in \mathbb{R}^{L_s \times C_{out}}$. Thus, the scalar form of $J_3$ can be expressed as

$$J_3 = 2\sum_{i=1}^{L_s}\sum_{j=1}^{C_{out}} A_{ij}C_{ij} = 2\sum_{i=1}^{L_s}\sum_{j=1}^{C_{out}}\sum_{k=1}^{C_{in}} X_{ik}^l \left(W_{kj}^l - \widehat{W}_{kj}^l M_{kj}^l\right)\sum_{r=1}^{C_{in}} X_{ir}^l B_{rj}^l M_{rj}^l$$

$$= 2\sum_{i,j}\sum_{k,r} X_{ik}^l X_{ir}^l \left(W_{kj}^l - \widehat{W}_{kj}^l M_{kj}^l\right)\left(B_{rj}^l M_{rj}^l\right), \tag{6}$$

Based on the statistical properties of $B_{rj}^l$, we have

$$\mathbb{E}[J_3] = 2\sum_{i,j}\sum_{k,r} X_{ik}^l X_{ir}^l (W_{kj}^l - \widehat{W}_{kj}^l M_{kj}^l)(\mathbb{E}[B_{rj}^l]M_{rj}^l) = 0, \tag{7}$$

By combining equation 5 with the constraint in equation 1, we obtain:

$$\underset{\mathbf{M}^l}{\arg\min} \left\|\mathbf{X}^l\mathbf{W}^l - \mathbf{X}^l(\widehat{\mathbf{W}}^l \odot \mathbf{M}^l)\right\|_F^2 + \left\|\mathbf{X}^l(\mathbf{B}^l \odot \mathbf{M}^l)\right\|_F^2 \quad s.t. \quad 1 - \frac{1}{N}\|\mathbf{M}^l\|_0 = p. \tag{8}$$

The above objective applies a uniform pruning ratio $p$ across all layers, ignoring the inherent differences among them.

### 3.4 BLOCK-WISE POST-QUANTIZATION PRUNING

Each layer in a Transformer block may have a different impact on the model's performance. If the model can adaptively adjust the pruning rates of individual layers while maintaining the overall pruning rate, it would enhance the performance of the pruned model. To achieve this, inspired by Xu et al. (2024), we modify the pruning approach from layer-wise to block-level pruning. Let $\mathbf{X}$ denotes the input to a Transformer block, which is processed through multiple neural network layers with the corresponding collection of parameters denoted by $\mathbf{W} := \{\mathbf{W}^l\}_{l=1}^L$ or $\widehat{\mathbf{W}} := \{\widehat{\mathbf{W}}^l\}_{l=1}^L$. The output of the block is given by $T(\mathbf{X}, \mathbf{W})$ or $T(\mathbf{X}, \widehat{\mathbf{W}})$. Accordingly, the loss function for block-level pruning is defined as:

$$\underset{\mathbf{M}^1,\ldots,\mathbf{M}^L}{\arg\min} \mathcal{L} := \|T(\mathbf{X}, \mathbf{W}) - T(\mathbf{X}, \widehat{\mathbf{W}} \odot \mathbf{M})\|_F^2 + \|T(\mathbf{X}, \mathbf{B} \odot \mathbf{M})\|_F^2 + \theta(1 - \frac{1}{S}\sum_{l=0}^L \|\mathbf{M}^l\|_0 - p)^2, \tag{9}$$

where $\mathbf{M}^l$ denotes the learnable binary mask corresponding to each $\widehat{\mathbf{W}}^l$, $\mathbf{M} := \{\mathbf{M}^l\}_{l=1}^L$ represents the collection of all such masks, $\mathbf{B} := \{\mathbf{B}^l\}_{l=1}^L$ denotes the collection of quantization error matrix, and $S$ denotes the total number of elements across all $\widehat{\mathbf{W}}^l$ matrices. $\theta$ is a trade-off hyperparameter that converts the constraint on the pruning rate into a regularization term.

The loss function consists of three components. The first term measures the discrepancy between the output of the original Transformer block and that of the post-quantization pruning Transformer block. The second term captures the influence of quantization error on pruning, while the third term ensures that the block-level pruning rate meets the desired target. This formulation yields a pruning mask that retains weights that are both important within the Transformer block and less affected by quantization errors.

**Mask Generation**: Next, we discuss how to allocate pruning rates within a Transformer block dynamically. To achieve this, we adopt the approach proposed in Xu et al. (2024), leveraging learnable sparsity to implement a differentiable binary mask. Specifically, we define a set of candidate pruning rates $\{r_d\}_{d=1}^D$ within the range $[0, 1)$, where each $r_d$ increases monotonically and $D$ is the

total number of candidates (e.g., $\{r_d\}_{d=1}^D = [0, 0.01, 0.02, \ldots, 0.99]$). Each candidate pruning rate is associated with a learnable weight parameter $\{\beta_1, \beta_2, \ldots, \beta_D\}$, which represents its selection probability. A softmax function is applied to ensure that all probabilities are positive and sum to one. Finally, we define the differentiable sparsity (pruning ratio) for the $i$-th row of $\widehat{\mathbf{W}}^l$ in a block as:

$$\alpha_{row}^l = \sum_{d=1}^D \beta_d r_d, \tag{10}$$

where $\alpha_{row}^l \in [0, 1]$ represents the learnable sparsity corresponding to each row of $\widehat{\mathbf{W}}^l$. Although the aforementioned definition of differentiable sparsity can be applied at the layer level, our experimental results (see Table 2) demonstrate that assigning a distinct sparsity ratio to each row within every layer leads to further improvements in performance. Therefore, we adopt this finer-grained sparsity allocation strategy.

To obtain the final binary pruning mask based on row sparsity, we first rank the elements of each row in $\widehat{\mathbf{W}}^l$ using the PQP metric in equation 2. Thus, for any $i$-th row $\widehat{\mathbf{W}}_{i,:}^l$, the corresponding importance scores of all elements are denoted by $\mathbf{I}_{i,:}^l$. In descending order of scores, we have:

$$\widehat{W}_{i,\hat{j}}^l = \text{sort}(\widehat{W}_{i,j}^l \mid I_{i,j}^l). \tag{11}$$

After that, row sparsity is achieved by retaining the top $K$ most important weights in each row, where $K = (1 - \alpha_{row}^l) \times C_{out}$. We then prune the unimportant weights by setting a row-wise binary mask $\mathbf{M}_{i,:} \in \{0, 1\}^{C_{out}}$ for each pruned row. The binary mask $\mathbf{M}_{i,j}$ is obtained by

$$\mathbf{M}_{i,j} = \begin{cases} 1 & I_{i,j} \text{ in top-}K \\ 0 & otherwise \end{cases}, \tag{12}$$

However, generating binary masks through the above hard thresholding impedes gradient computation during backpropagation. To overcome this, we redefine the pruning probability for each element in $\widehat{\mathbf{W}}_{i,:}^l$ as follows:

$$P(\widehat{W}_{i,\hat{j}}) = \sum_{d=k+1}^D \beta_d, \quad \text{if } r_k \leq \frac{\hat{j}}{C_{out}} < r_{k+1}, \tag{13}$$

where $r_k$ and $r_{k+1}$ are two adjacent elements in $\{r_d\}_{d=1}^D$. We set $r_0 = 0$ and $\beta_D = 0$ to ensure that the most important weights are always maintained. Based on this, we reformulate equation 12 as follows:

$$\mathbf{M}_{i,j} = \begin{cases} 1 & P(\widehat{W}_{i,\hat{j}}) \geq \alpha_{row}^l \\ 0 & otherwise \end{cases}. \tag{14}$$

**Differentiability of Sparsity** $\alpha_{row}^l$**:** It is clear that the loss in equation 9 is a function of the variable $\mathbf{M}_{i,j}$. According to equation 13 and equation 14, $\mathbf{M}_{i,j}$ can be reformulated as a function of $\beta_i$. Furthermore, based on equation 10, $\alpha_{row}$ can also be expressed as a function of $\beta_i$. Accordingly, the gradient of the loss function $\mathcal{L}$ with respect to $\alpha_{row}^l$ can be written as:

$$\frac{\partial \mathcal{L}}{\partial \alpha_{row}^l} = r_i \sum_{i=1}^n \frac{\partial \mathcal{L}}{\partial \beta_i}. \tag{15}$$

Subsequently, the gradient of $\mathcal{L}$ with respect to $\beta_i$ can be derived as:

$$\frac{\partial \mathcal{L}}{\partial \beta_i} = \sum_{j=1}^{C_{out}} \frac{\partial \mathcal{L}}{\partial \mathbf{M}_{i,j}} \frac{\partial \mathbf{M}_{i,j}}{\partial P(\widehat{W}_{i,\hat{j}})} \frac{\partial P(\widehat{W}_{i,\hat{j}})}{\partial \beta_i}. \tag{16}$$

Note that the gradient of mask $\mathbf{M}_{i,j}$ with respect to pruning probability $P(\widehat{W}_{i,\hat{j}})$ can be estimated using Straight-Through-Estimator (STE)(Yin et al., 2019). Through equation 15 and equation 16, the row-wise sparsity (pruning rate) can be optimized through a simple gradient descent algorithm. The complete PQP algorithm is presented in Appendix E.

Table 2: WikiText perplexity results for LLaMA and LLaMA-2 models following 4-bit GPTQ quantization and pruning at a 50% sparsity level. The best results are highlighted in **bold**.

| Method | Sparsity | LLaMA | | | | LLaMA-2 | | |
|---|---|---|---|---|---|---|---|---|
| | | 7B | 13B | 30B | 65B | 7B | 13B | 70B |
| Dense | - | 5.68 | 5.09 | 4.10 | 3.53 | 5.47 | 4.88 | 3.32 |
| SparseGPT (Frantar & Alistarh, 2023) | 50% | 9.21 | 6.50 | 5.55 | 4.84 | 7.72 | 6.29 | 4.48 |
| Wanda (Sun et al., 2024) | 50% | 7.98 | 6.58 | 5.60 | 4.92 | 7.53 | 6.40 | 4.58 |
| RIA (Zhang et al., 2024b) | 50% | 8.21 | 6.60 | 5.55 | 4.91 | 7.67 | 6.35 | 4.61 |
| BESA (Xu et al., 2024) | 50% | 7.38 | 6.40 | 5.34 | 4.67 | 7.55 | 6.19 | 4.47 |
| BaWA (Liu et al., 2025b) | 50% | 8.10 | 6.65 | 5.55 | 4.91 | 7.58 | 6.33 | 4.59 |
| PQP (layer-wise) | 50% | 7.67 | 6.31 | 5.30 | 4.61 | 7.17 | 6.03 | 4.29 |
| PQP (row-wise) | 50% | **7.08** | **6.07** | **5.10** | **4.45** | **6.89** | **5.84** | **4.23** |

# 4 EXPERIMENTS

To evaluate the effectiveness of our model compression method, we conduct experiments on two widely used LLMs and further perform ablation studies to gain deeper insights into the PQP method.

## 4.1 EXPERIMENTAL SETUP

**Models and Evaluation:** In this section, we conduct extensive experiments on multiple baseline models to evaluate the proposed method. The baseline models are selected following prior works (Frantar & Alistarh, 2023; Sun et al., 2024), specifically including LLaMA 7B/13B/30B/65B (Touvron et al., 2023a) and LLaMA-2 7B/13B/70B (Touvron et al., 2023b). All models are loaded using the HuggingFace Transformers library (Wolf et al., 2020) and executed on two NVIDIA A100 GPUs (80GB). Post-quantization pruned models are evaluated using perplexity and zero-shot accuracy. Perplexity is measured on the WikiText2 validation set (Merity et al., 2016), with lower values indicating better performance. Zero-shot accuracy is assessed using the EleutherAI LM Harness (Gao et al., 2021) across seven tasks: PIQA (Bisk et al., 2020), BoolQ (Clark et al., 2019), HellaSwag (Zellers et al., 2019), ARC-Easy, ARC-Challenge (Clark et al., 2018), and OpenbookQA (Mihaylov et al., 2018), where higher values indicate better performance.

**Implementation details:** The calibration dataset, sampled from the C4 dataset (Raffel et al., 2020), comprises 128 sequences, each containing 2048 tokens. We conduct post-quantization pruning experiments on LLaMA models and LLaMA-2 models under a consistent experimental environment and codebase. For the quantization process, we uniformly apply the 4-bit GPTQ quantization (Frantar et al., 2023). (The impact of different quantization methods on model performance is detailed in Appendix J.) The subsequent block-wise pruning experiments follow the configuration outlined in (Xu et al., 2024).

**Baseline.** We compare our PQP method with existing pruning techniques as follows: **1)** SparseGPT (Frantar & Alistarh, 2023): pruning using second-order information from Taylor expansion; **2)** Wanda (Sun et al., 2024): pruning based on the magnitude of weights and input activations; **3)** RIA (Zhang et al., 2024b): pruning based on the relative importance of weight magnitude and the square root of input activations; **4)** BESA (Xu et al., 2024): pruning with block-wise dynamic sparsity based on the metric in Wanda; **5)** BaWA (Liu et al., 2025b): (Liu et al., 2025b) introduces a pruning criterion based on the balance between weight and activation distributions, with the balancing factors set as $\theta_1 = 1$, $\theta_2 = 1$, and $\theta_3 = 0.5$, following the original configuration.

## 4.2 MAIN RESULTS

In the following, we present the changes in perplexity before and after model compression, along with the corresponding variations in zero-shot tasks.

**Perplexity Experiments**: This study reports on the perplexity of PQP models under 50% pruning rate. All comparisons are conducted on 4-bit post-quantization models to ensure consistency. As shown in Table 2, PQP consistently achieves the lowest perplexity across LLaMA-7B to LLaMA-65B and LLaMA-2-7B to LLaMA-2-70B. For example, PQP reduces perplexity by 0.30 on LLaMA-7B, 0.22 on LLaMA-65B, 0.66 on LLaMA-2-7B, and 0.24 on LLaMA-2-70B compared to the next-best method. These results demonstrate PQP's robustness and effectiveness across different models.

Table 3: Average zero-shot accuracies (%) of post-quantization pruning LLaMA and LLaMA-2 models on the PIQA, BoolQ, HellaSwag, WinoGrande, ARC, and OBQA datasets. The best-performing results are indicated in **bold**.

| Method | Sparsity | LLaMA | | | | LLaMA-2 | | |
|---|---|---|---|---|---|---|---|---|
| | | 7B | 13B | 30B | 65B | 7B | 13B | 70B |
| Dense | - | 61.84 | 63.80 | 67.41 | 68.57 | 61.77 | 64.92 | 69.06 |
| SparseGPT (Frantar & Alistarh, 2023) | 50% | 52.48 | 59.78 | 64.81 | 67.01 | 56.92 | 58.09 | 67.88 |
| Wanda (Sun et al., 2024) | 50% | 56.01 | 59.97 | 64.24 | 66.66 | 56.92 | 61.26 | 67.09 |
| RIA (Zhang et al., 2024b) | 50% | 55.66 | 58.10 | 64.07 | 66.13 | 56.22 | 60.66 | 66.99 |
| BESA (Xu et al., 2024) | 50% | 57.39 | 60.95 | 65.40 | 67.63 | 58.04 | 62.14 | 68.20 |
| BaWA (Liu et al., 2025b) | 50% | 55.64 | 60.15 | 63.22 | 66.27 | 55.73 | 61.01 | 66.99 |
| PQP (our) | 50% | **57.66** | **61.52** | **65.57** | **68.12** | **58.54** | **62.41** | **68.33** |

Table 4: Evaluation of post-quantization pruning metrics at 50% unstructured layer-wise sparsity and semi-structured 2:4 and 4:8 sparsity patterns, using perplexity on the WikiText2 dataset.

| Method | Sparsity | LLaMA | | | | LLaMA-2 | | |
|---|---|---|---|---|---|---|---|---|
| | | 7B | 13B | 30B | 65B | 7B | 13B | 70B |
| Dense | - | 5.68 | 5.09 | 4.10 | 3.53 | 5.47 | 4.88 | 3.32 |
| SparseGPT Frantar & Alistarh (2023) | 50% | 9.21 | 6.50 | 5.55 | 4.84 | 7.72 | 6.29 | 4.48 |
| Wanda Sun et al. (2024) | 50% | 7.98 | 6.58 | 5.60 | 4.92 | 7.53 | 6.40 | 4.58 |
| RIA Zhang et al. (2024b) | 50% | 8.21 | 6.60 | 5.55 | 4.91 | 7.67 | 6.35 | 4.61 |
| BaWA (Liu et al., 2025b) | 50% | 8.10 | 6.65 | 5.55 | 4.91 | 7.58 | 6.33 | 4.59 |
| PQP-Metric (ours) | 50% | **7.60** | **6.36** | **5.36** | **4.65** | **7.22** | **6.06** | **4.34** |
| SparseGPT Frantar & Alistarh (2023) | 4:8 | 11.11 | 7.69 | 6.35 | 5.63 | 10.23 | 7.32 | 5.14 |
| Wanda Sun et al. (2024) | 4:8 | 10.34 | 8.33 | 6.59 | 5.84 | 10.10 | 7.96 | 5.28 |
| RIA Zhang et al. (2024b) | 4:8 | 10.65 | 8.18 | 6.54 | 5.79 | 10.83 | 7.76 | 5.24 |
| BaWA (Liu et al., 2025b) | 4:8 | 10.19 | 8.14 | 6.52 | 5.80 | 9.86 | 7.65 | 5.20 |
| PQP-Metric (ours) | 4:8 | **8.94** | **7.35** | **6.04** | **5.33** | **8.94** | **6.92** | **4.85** |
| SparseGPT Frantar & Alistarh (2023) | 2:4 | 15.43 | 9.30 | 7.54 | 6.72 | 13.29 | 9.24 | 5.94 |
| Wanda Sun et al. (2024) | 2:4 | 15.01 | 11.03 | 7.90 | 6.88 | 14.93 | 10.61 | 6.13 |
| RIA Zhang et al. (2024b) | 2:4 | 14.59 | 10.72 | 7.85 | 7.05 | 14.98 | 9.62 | 6.13 |
| BaWA (Liu et al., 2025b) | 2:4 | 14.21 | 10.30 | 7.84 | 6.89 | 14.01 | 9.70 | 5.99 |
| PQP-Metric (ours) | 2:4 | **12.95** | **9.13** | **7.02** | **6.26** | **13.11** | **8.57** | **5.52** |

**Zero-shot Tasks**: To further evaluate the generalizability of our approach, we conduct experiments on seven zero-shot tasks. Due to significant performance variations of each algorithm across different tasks, we use the average accuracy over all seven tasks as the primary evaluation metric. The results in Tables 3 show that, across all models, our method consistently achieves the highest average zero-shot accuracy, demonstrating strong generalizability and robustness.

## 4.3 ABLATION STUDY

**Pruning Metric Selection:** To validate the effectiveness of the pruning metric proposed in equation 2, we design experiments to evaluate model performance under various one-shot pruning metrics. To minimize the influence of confounding factors and enhance the reliability of our analysis, we evaluate the pruning metrics under three fixed sparsity settings: layer-wise unstructured pruning at 50% sparsity, and semi-structured 2:4 and 4:8 sparsity patterns. As shown in Table 4, PQP-Metric consistently achieves lower perplexity than SparseGPT, Wanda, RIA, and BaWA across both unstructured and semi-structured sparsity regimes, demonstrating its effectiveness and strong generalizability for one-shot post-quantization pruning.

Table 5: Impact of the choice of $PM(\cdot)$ in PQP-Metric.

| $PM(\cdot)$ | LLaMA | | LLaMA-2 | |
|---|---|---|---|---|
| | 1-7B | 1-13B | 2-7B | 2-13B |
| Wanda | 7.84 | 6.43 | 7.30 | 6.13 |
| BaWA | 7.60 | **6.34** | 7.25 | 6.09 |
| RIA | **7.60** | 6.36 | **7.22** | **6.06** |

Additionally, we conduct an ablation study on the pruning metric $PM(\mathbf{W})$ used in PQP-Metric. For comparison, we choose Wanda, RIA, and BaWA, as all three avoid weight reconstruction and gradient computation. As shown in Table 5, integrating RIA into our method further improves performance.

**Loss Function Analysis:** Here, we conduct comparative experiments to evaluate the impact of our improved loss function equation 9 on model performance after post-quantization pruning. First, we compare the full loss function equation 9 with a variant that removes the second term related to quantization error. Experiments are performed on LLaMA-1 and LLaMA-2 models under 3-bit and 4-bit quantization. As shown in Table 6, incorporating quantization error into the loss function consistently improves performance across different settings.

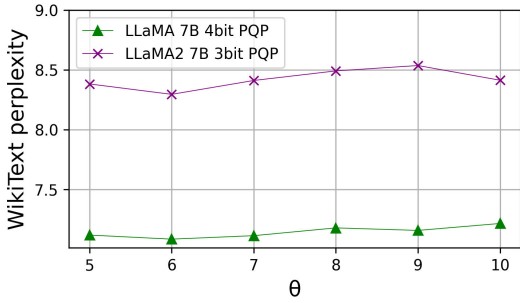

Figure 2: WikiText perplexity under different hyperparameter $\theta$ in the LLaMA-7B and LLaMA-2-7B models.

We also analyze the effect of the hyperparameter $\theta$ in equation 9. As shown in Figure 2, model performance is relatively stable across different values of $\theta$, with $\theta = 6$ achieving the best results. We therefore use this setting in all experiments.

**Computing Efficiency Analysis:** PQP adopts a one-shot quantization and one-shot pruning strategy, with the main computational overhead arising from the block-wise pruning stage. As shown in equation 13, equation 15, and equation 16, PQP introduces $D$ candidate pruning rates and implements a learnable, differentiable pruning rate via their weighted sum. Specifically, each $\alpha_{\text{row}}^l$ is associated with $D$ learnable weight parameters. The value of $D$ is controllable and typically small; in our experiments,

Table 6: Loss function comparison on LLaMA-1-7B (1-7B) and LLaMA-2-7B (2-7B) under different quantization bits.

| Method | Wikitext2 | C4 |
|---|---|---|
| 1-7B 4bit PQP (variant loss) | 7.20 | 9.21 |
| 1-7B 4bit PQP (loss in (9)) | **7.08** | **9.20** |
| 2-7B 3bit PQP (variant loss) | 8.40 | 10.89 |
| 2-7B 3bit PQP (loss in (9)) | **8.30** | **10.84** |

we set $D = 100$, which results in $100 \times C_{\text{in}}$ additional learnable parameters per layer. To further reduce parameter overhead, we propose a layer-wise learning strategy, where all rows within a layer share the same pruning rate. This reduces the number of learnable parameters per layer to $D$. As shown in Table 2, although this approach significantly reduces overhead, it results in slightly degraded performance compared to the row-wise strategy. Overall, PQP introduces only a minimal number of additional trainable parameters—approximately 1.76% of those in a single Transformer block for row-wise learning, and just 0.0003% for layer-wise learning.

Updating the learnable pruning rates requires only a small amount of calibration data (see Appendix F). Similar to Xu et al. (2024), this process is highly efficient and controllable. In general, a single epoch of training on the calibration set is sufficient to update the pruning rates (see Appendix G).

## 5 CONCLUSION

For large-scale model compression, this paper presents a novel, low-cost, and efficient post-quantization pruning (PQP) method. As a one-shot strategy, PQP achieves high compression ratios with minimal computational overhead and performance degradation by tightly integrating quantization and pruning. We propose a new parameter importance metric tailored for post-quantization scenarios, which combines information from both original and quantized weights to more accurately guide pruning. Additionally, we integrate the statistical characteristics of quantization error into the block-wise reconstruction loss, allowing the pruning process to retain weights that are important and less impacted by quantization noise while discarding others. The resulting compression pipeline is lightweight, effective, easy to implement, and requires no additional fine-tuning. Experimental results on LLaMA and LLaMA-2 models demonstrate that PQP consistently outperforms existing one-shot compression methods across a wide range of models.

However, our findings also reveal that when the overall compression ratio exceeds $1/8$, performance degradation becomes more noticeable. Addressing this limitation in a low-cost, one-shot framework remains an important direction for future work.

ETHICS STATEMENT

This work focuses on model compression techniques, including pruning and quantization, to improve the efficiency of large language model inference. Our research does not involve human subjects or the collection of sensitive personal data. All datasets used are publicly available and widely adopted in prior research, and we comply with their corresponding licenses. The goal of this work is to advance efficient and sustainable AI research by reducing computational and energy costs during model deployment.

We acknowledge that large language models may generate biased or harmful content if misused. However, our contributions are methodological and orthogonal to such risks. We encourage responsible use of compressed models and adherence to ethical guidelines in downstream applications. We confirm compliance with the ICLR Code of Ethics.

REPRODUCIBILITY STATEMENT

We have made every effort to ensure the reproducibility of our results. All datasets used in this study are publicly available existing datasets, and no private or proprietary data were involved. The experimental setup, including training procedures, model configurations, and hardware specifications, is described in detail in the paper and supplementary materials. To further facilitate replication and verification, the implementation code for the proposed algorithm is provided as an anonymously downloadable repository.

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
