## A   USE OF LLMs

Large Language Models (LLMs) were used to assist in the writing and polishing of this manuscript. Specifically, we employed an LLM to refine the language, improve readability, and enhance clarity across various sections of the paper. The model supported tasks such as sentence rephrasing, grammar checking, and improving the overall flow of the text.

It is important to emphasize that the LLM was not involved in the ideation, research methodology, or experimental design. All research concepts, ideas, and analyses were entirely developed and conducted by the authors. The contributions of the LLM were limited to improving the linguistic quality of the manuscript, without influencing the scientific content or data analysis.

The authors take full responsibility for the content of the manuscript, including any text generated or refined with LLM assistance. We have ensured that the use of LLMs adheres to ethical guidelines and does not contribute to plagiarism or scientific misconduct.

## B   POST-TRAINING QUANTIZATION WITH GPTQ

In the post-training quantization method GPTQ (Frantar et al., 2023), the weight matrix columns are grouped. Within each group, weights are quantized one column at a time. After quantization, the unquantized weights in other columns are compensated for the error introduced by quantization to reduce the accuracy loss. Once all weights in a group are quantized, the group compensates for unquantized weights in other groups. This process continues until all weights in the matrix are quantized. The error introduced by weight quantization and the error compensation are calculated using the method from OBQ (Frantar & Alistarh (2022)). The error $w_q$ caused by quantized weights and the error compensation $\delta$ for the remaining unquantized weights within the group are calculated as follows:

$$w_q = \mathrm{argmin}_{w_q} \frac{(\mathrm{quant}(w_q) - w_q)^2}{[\mathbf{H}_l^{-1}]_{qq}},$$
$$\delta = -\frac{w_q - \mathrm{quant}(w_q)}{[\mathbf{H}_l^{-1}]_{qq}} \cdot (\mathbf{H}_l^{-1})_{:,q}. \tag{17}$$

where the quantization operation, $quant(\cdot)$, uses simple rounding, $q$ is the column index in the weight matrix, the second-order layer-wise Hessian matrix is $\mathbf{H}_l = \mathbf{X}_l^\top \mathbf{X}_l$, and $\mathbf{H}_l^{-1}$ is computed using Cholesky decomposition. This improves numerical stability and removes the need to update the Hessian matrix.

## C   PQP METRIC WITH GRADIENT INFORMATION

Inspired by Pruner-zero Dong et al. (2024), we enhance post-quantization pruning by incorporating gradient information into the pruning metric to improve model performance. Specifically, we define:

$$PM(W) = |\mathbf{W}| * |\mathbf{W}| * \mathrm{mms}(|\mathbf{G}|). \tag{18}$$

As shown in Table 7, we present the model performance under our PQP that includes gradient information. We also compare this with the performance of other post-quantization pruning methods using the one-shot pruning metric. The results show that the post-quantization pruned model with gradient information achieves a lower perplexity. However, since gradient information requires more computational resources, it is difficult to obtain gradient information for larger models under limited resources. Our experiments are only conducted on the LLaMA-13B and LLaMA-2-13B models. Gradient information clearly provides a performance advantage in model compression; however, this benefit comes with a trade-off between computational cost and efficiency.

## D   COMPARISON OF PQP-METRIC WITH ORIGINAL-WEIGHT-BASED PRUNING METRICS

All comparison methods in Table 2 assess weight importance using post-quantization pruned weights. To further validate the effectiveness of our proposed method, we also compare PQP-Metric with

Table 7: Effect of incorporating gradient information into $PM(\cdot)$ on WikiText perplexity, where "G" indicates that gradient information is included in the pruning metric.

| Method | Sparsity | LLaMA-7B | LLaMA-13B | LLaMA-2-7B | LLaMA-2-13B |
|---|---|---|---|---|---|
| Dense | - | 5.68 | 5.09 | 5.47 | 4.88 |
| Pruner-Zero Dong et al. (2024) | 50% | 7.77 | 6.46 | 7.45 | 6.16 |
| PQP-Metric (layer-wise) | 50% | 7.60 | 6.36 | 7.22 | 6.06 |
| PQP-Metric-G (layer-wise) | 50% | **7.40** | **6.22** | **7.17** | **6.04** |
| PQP-Metric (block-wise) | 50% | 7.08 | 6.07 | 6.89 | 5.84 |
| PQP-Metric-G (block-wise) | 50% | **7.00** | **6.03** | **6.78** | **5.80** |

pruning metrics based on the original (pre-quantization) weights. Specifically, we perform 3-bit quantization on the LLaMA and LLaMA-2 models and apply pruning metrics derived from the original weights $W$ to evaluate weight importance in the quantized model. The pruned models are then evaluated on both the WikiText2 dataset and seven zero-shot tasks. Perplexity results on WikiText2 are presented in Table 8, and the average zero-shot accuracy across the seven tasks is shown in Table 9. As demonstrated in both tables, our PQP-Metric consistently outperforms the weight-based pruning baselines across all evaluations.

Table 8: Comparison of PQP-metric and original-weight-based pruning metrics on WikiText perplexity.

| Method | Sparsity | LLaMA-7B | LLaMA-13B | LLaMA-2-7B | LLaMA-2-13B |
|---|---|---|---|---|---|
| Dense | - | 5.68 | 5.09 | 5.47 | 4.88 |
| SparseGPT | 50% | 32.41 | 7.37 | 10.86 | 7.23 |
| Wanda | 50% | 9.46 | 7.21 | 9.53 | 7.29 |
| RIA | 50% | 9.26 | 7.15 | 9.27 | 6.97 |
| BESA | 50% | 8.74 | 7.01 | 9.12 | 6.88 |
| BaWA | 50% | 9.00 | 7.15 | 9.12 | 6.99 |
| PQP (ours) | 50% | **8.21** | **6.72** | **8.44** | **6.58** |

Table 9: Comparison of PQP-metric and original-weight-based pruning metrics on average accuracy across 7 zero-shot tasks in the PIQA, BoolQ, HellaSwag, WinoGrande, ARC, and OBQA Datasets.

| Method | Sparsity | LLaMA-7B | LLaMA-13B | LLaMA-2-7B | LLaMA-2-13B |
|---|---|---|---|---|---|
| Dense | - | 61.84 | 63.80 | 61.77 | 64.92 |
| SparseGPT | 50% | 36.88 | 56.77 | 52.77 | 58.67 |
| Wanda | 50% | 54.52 | 57.20 | 54.73 | 59.79 |
| RIA | 50% | 54.23 | 57.30 | 53.36 | 59.11 |
| BESA | 50% | 55.55 | 58.22 | 53.99 | 58.80 |
| BaWA | 50% | 54.78 | 57.02 | 53.50 | 58.40 |
| PQP (ours) | 50% | **55.73** | **58.37** | **54.81** | **59.80** |

# E  PSEUDOCODE OF BLOCK-WISE PQP

# F  ABLATION OF CALIBRATION SIZE

We investigate the impact of calibration dataset size on the performance of PQP and other pruning importance metrics that require calibration data. As shown in Figure 3, PQP consistently maintains robust performance as the number of calibration samples increases. Notably, across all dataset sizes, our method consistently outperforms other baselines in terms of perplexity, further demonstrating the robustness of our approach.

---

**Algorithm 1** Pseudocode of Block-wise PQP.

---

**Input**: Calibration data $\mathbf{X}$, pre-trained model $\{\mathbf{W}^l\}_{l=1}^L$ and block-wise target sparsity $p$.
**Output**: Post-quantization pruned model.
 1: **for** $l \in \{1, 2, \ldots, L\}$ **do**
 2:     $\widehat{\mathbf{W}}^l \leftarrow$ quantization by GPTQ;
 3:     $\mathbf{B}^l \leftarrow$ calculate quantization error;
 4:     Sort $\widehat{\mathbf{W}}^l$ by the metric in (2);
 5:     **for** $\widehat{\mathbf{W}}_{i,:}^l \in \widehat{\mathbf{W}}^l$ **do**
 6:         **while** sparsity $\alpha_{row}^l$ not converge **do**
 7:             $\alpha_{row}^l \leftarrow$ initialize learnable sparsity via (10);
 8:             $\mathbf{M}_{i,j}^l \leftarrow$ initialize element-wise binary mask via (11) - (14);
 9:             $\mathcal{L} \leftarrow$ calculate loss function via (9);
10:             Update $\alpha_{row}^l$ and $\{\beta_d\}_{d=1}^D$ using back-propagation via (15) and (16);
11:         **end while**
12:     **end for**
13:     $\mathbf{M}^l \leftarrow$ generate the quantized weights mask.
14: **end for**

---

Additionally, both our method and BESA require training on the calibration dataset to enable learnable pruning rate adjustment. The training is conducted for one epoch with a batch size of 1—i.e., one sample is introduced at a time to update the learnable pruning rates. During this process, we save the generated mask matrices after incorporating [1, 2, 4, 8, 16, 32, 64, 128, 256] samples and evaluate the model's performance after pruning with these masks. The red and blue curves in Figure 3 reflect these results, offering insights into the convergence behavior and stability of PQP and BESA during the learnable pruning rate training process.

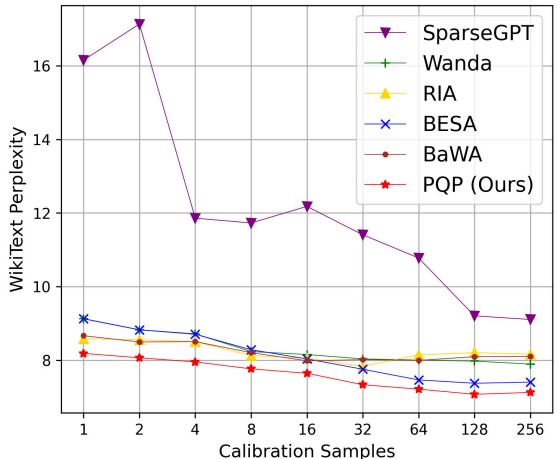

Figure 3: Ablation of calibration data size in LLaMA-7B.

## G INFLUENCE OF TRAINING EPOCHS

We examine the impact of training epochs on our post-quantization pruning method using calibration data from the C4 dataset. The detailed performance results are presented in Table 10. Notably, the perplexity of the pruned LLaMA-7B model on both WikiText2 and C4 increases as the number of training epochs grows, indicating that additional training does not necessarily lead to better pruning performance. Therefore, we adopt a single training epoch as the default setting.

Table 10: Ablation Study on training epochs for the LLaMA-7B model.

| Epochs | 1 | 3 | 5 | 7 | 10 |
|---|---|---|---|---|---|
| Wikitext2 | 7.08 | 7.12 | 7.19 | 7.18 | 7.13 |
| C4 | 9.20 | 9.20 | 9.20 | 9.20 | 9.22 |

## H  VARYING SPARSITY LEVELS

We also conduct experiments on different levels of unstructured pruning in LLaMA-7B, as shown in Figure 4. As the sparsity increases, PQP consistently outperforms all other methods in terms of model performance.

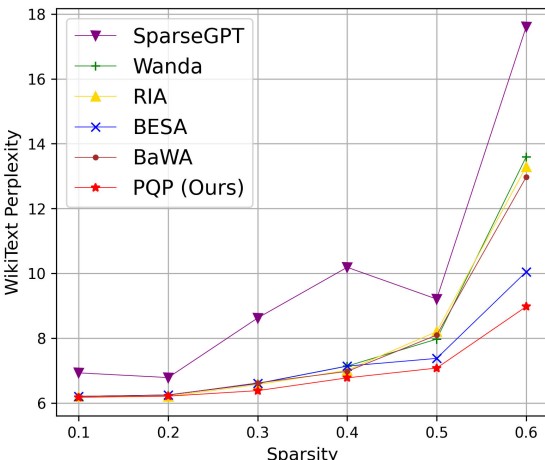

Figure 4:  Ablation of sparsity in LLaMA-7B.

## I  IMPACT OF QUANTIZATION BITS

To validate the effectiveness of our method in mitigating quantization error, we evaluate its performance across different bit-widths, corresponding to varying levels of quantization error. As shown in Figure 5, for the smaller LLaMA-13B model with fewer parameters, the quantization error at 8-bit precision is relatively low, making the advantage of our method less pronounced. However, as the bit-width decreases and the quantization error increases, our method demonstrates increasingly significant benefits. In the case of the larger LLaMA-65B model, our method shows clear advantages even at higher bit-widths, and its effectiveness further improves as bit-width decreases. These results confirm that the proposed PQP method effectively mitigates the impact of quantization error on model performance, with its benefits becoming more pronounced under more severe quantization conditions.

## J  GENERALITY ACROSS DIFFERENT QUANTIZATION METHODS

In the post-quantization pruning experiments presented in this paper, GPTQ is consistently employed as the quantization method. To demonstrate that our method's effectiveness is not dependent on the choice of quantization technique, we replace GPTQ with the simple and widely used Round To Nearest (RTN) method, which does not incorporate error compensation for quantized weights. All methods are evaluated under identical experimental conditions. As shown in Table 11, our method continues to deliver the most effective pruning performance compared to other approaches on the LLaMA-1 and LLaMA-2 series models under 4-bit RTN quantization. This further validates the generalizability of our PQP approach across different quantization methods and underscores that the core strength of our method lies in optimizing the quantization error introduced during quantization.

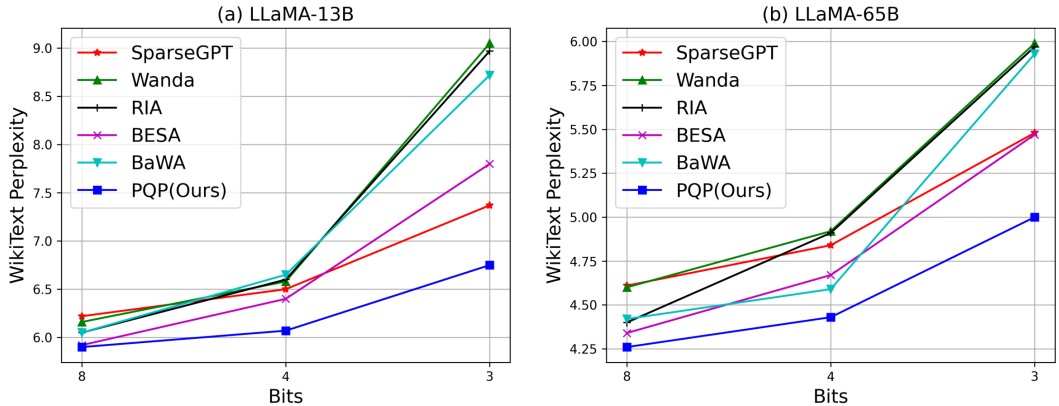

Figure 5: Effect of quantization bit-width on model performance. We use GPTQ for quantization and evaluate using WikiText perplexity. (a) shows results for LLaMA-13B, and (b) for LLaMA-65B.

Table 11: Post-quantization pruning for LLaMA and LLaMA-2 models with unstructured 50% sparsity and 4-bit Round To Nearest(RTN) quantization. The **best** performing result is indicated in **bold**.

| Quantization | Methods | Sparsity | LLaMA | | | | LLaMA-2 | | |
|---|---|---|---|---|---|---|---|---|---|
| | | | 7B | 13B | 30B | 65B | 7B | 13B | 70B |
| | Dense | - | 5.68 | 5.09 | 4.10 | 3.53 | 5.47 | 4.88 | 3.32 |
| | SparseGPT | 50% | 8.39 | 6.90 | 5.93 | 5.03 | 7.96 | 6.58 | 4.74 |
| | Wanda | 50% | 8.18 | 6.93 | 5.88 | 5.07 | 8.01 | 6.63 | 4.76 |
| RTN | RIA | 50% | 8.11 | 6.82 | 5.73 | 4.99 | 8.00 | 6.54 | 4.71 |
| | BESA | 50% | 7.59 | 6.59 | 5.62 | 4.80 | 7.51 | 6.31 | 4.57 |
| | BaWA | 50% | 8.09 | 6.83 | 5.72 | 4.97 | 8.00 | 6.50 | 4.69 |
| | PQP (ours) | 50% | **7.19** | **6.23** | **5.35** | **4.59** | **7.10** | **5.91** | **4.29** |

## K    PERPLEXITY ON THE C4 DATASET

To validate the generalizability of our method across different datasets, we compare the C4 perplexity of various approaches across different LLaMA model sizes. All methods are evaluated under consistent experimental settings, specifically 4-bit GPTQ quantization combined with 50% sparsity post-quantization pruning. As shown in Table 12, our post-quantization pruning method consistently achieves superior perplexity on the C4 dataset, further confirming its effectiveness.

Table 12: C4 perplexity results of post-quantization pruning for LLaMA and LLaMA-2 models using unstructured 50% sparsity and 4-bit GPTQ quantization. The **best** results are highlighted in **bold**.

| Datasets | Methods | Sparsity | LLaMA | | | | LLaMA-2 | | |
|---|---|---|---|---|---|---|---|---|---|
| | | | 7B | 13B | 30B | 65B | 7B | 13B | 70B |
| | Dense | - | 7.34 | 6.80 | 6.13 | 5.81 | 7.26 | 6.73 | 5.71 |
| | SparseGPT | 50% | 11.36 | 8.45 | 7.58 | 6.91 | 9.80 | 8.51 | 6.68 |
| | Wanda | 50% | 9.94 | 8.60 | 7.70 | 7.03 | 9.99 | 8.86 | 6.89 |
| C4 | RIA | 50% | 9.96 | 8.57 | 7.69 | 7.06 | 10.25 | 8.72 | 6.94 |
| | BESA | 50% | 9.55 | 8.39 | 7.50 | 6.83 | 9.61 | 8.48 | 6.75 |
| | BaWA | 50% | 9.92 | 8.54 | 7.70 | 7.03 | 10.11 | 8.68 | 6.92 |
| | PQP (ours) | 50% | **9.20** | **8.09** | **7.18** | **6.56** | **9.19** | **8.06** | **6.51** |

## L    DETAILED RESULTS OF ZERO-SHOT TASKS.

In the context of zero-shot learning, our evaluation encompasses a diverse set of tasks, as presented in Table 3. These tasks include PIQA(Bisk et al., 2020), BoolQ (Clark et al., 2019), HellaSwag (Zellers et al., 2019), ARC Easy and Challenge (Clark et al., 2018), and OpenbookQA (Mihaylov et al., 2018). To ensure the reproducibility of our results, we adhere to the settings and methodologies outlined in the BESA study (Xu et al., 2024). Detailed task-wise performance metrics are systematically presented in Tables 13 and 14. For certain specific tasks, our method does not yield the best performance. We attribute this to PQP's use of a generalized metric for assessing weight importance. While this metric performs well overall, it may not be optimal for tasks that rely on specialized knowledge pathways. In our evaluation setting, without task-specific fine-tuning, the models depend entirely on general knowledge learned during pretraining, which can lead to subpar performance on some individual tasks.

Table 13: Zero-shot accuracies (%) of post-quantization pruning LLaMA models on the PIQA, BoolQ, HellaSwag, WinoGrande, ARC, and OBQA datasets with unstructured 50% sparsity. 1-7/13/30/65B denotes LLaMA-7/13/30/65B. The **best** performing result is indicated in **bold**, while the second best is underlined.

| Models | Methods | PIQA | BoolQ | HellaSwag | WinoGrande | ARC-e | ARC-c | OBQA | Average |
|--------|---------|------|-------|-----------|------------|-------|-------|------|---------|
| | Dense | 78.67 | 75.63 | 56.95 | 70.09 | 75.25 | 41.89 | 34.40 | 61.84 |
| 1-7B | SparseGPT | 71.71 | 67.95 | 46.65 | 63.54 | 62.54 | 32.17 | 22.80 | 52.48 |
| | Wanda | 74.32 | 69.27 | 50.06 | 65.75 | 68.43 | 35.24 | 29.00 | 56.01 |
| | RIA | 74.54 | 68.99 | 49.52 | 66.61 | 67.76 | 34.81 | 27.40 | 55.66 |
| | BESA | **76.22** | 70.64 | 52.40 | **67.48** | 68.43 | 36.77 | 29.80 | 57.39 |
| | BaWA | 74.70 | 69.91 | 49.33 | 66.14 | 67.26 | 34.13 | 28.00 | 55.64 |
| | PQP(ours) | 76.12 | **71.07** | **52.96** | 66.77 | **69.44** | **37.03** | **30.20** | **57.66** |
| | Dense | 79.16 | 77.98 | 59.92 | 72.61 | 77.31 | 46.42 | 33.20 | 63.80 |
| 1-13B | SparseGPT | 77.42 | 75.38 | 54.02 | 69.93 | 71.00 | 39.51 | 31.20 | 59.78 |
| | Wanda | 76.33 | 76.42 | 54.50 | 69.85 | 71.21 | 41.89 | 29.60 | 59.97 |
| | RIA | 76.28 | 75.20 | 53.52 | 70.24 | 72.22 | 39.16 | 30.08 | 58.10 |
| | BESA | 77.04 | **76.73** | 56.18 | 69.69 | 73.11 | 41.89 | 32.00 | 60.95 |
| | BaWA | 76.82 | 75.11 | 53.73 | 70.40 | 73.15 | 40.44 | 31.40 | 60.15 |
| | PQP(ours) | **77.64** | 76.39 | **56.61** | **70.56** | **73.53** | **43.34** | **32.60** | **61.52** |
| | Dense | 81.01 | 82.35 | 63.34 | 75.93 | 80.39 | 52.82 | 36.00 | 67.41 |
| 1-30B | SparseGPT | 78.94 | **81.70** | 60.35 | **74.35** | 78.05 | 48.46 | 32.80 | 64.81 |
| | Wanda | 78.56 | 79.30 | 60.40 | 72.93 | 77.45 | 47.87 | 33.20 | 64.24 |
| | RIA | 78.40 | 81.40 | 59.20 | 72.45 | 77.05 | 45.99 | 34.00 | 64.07 |
| | BESA | **79.65** | 81.50 | 62.00 | 73.88 | 77.65 | 48.29 | 34.80 | 65.40 |
| | BaWA | 78.13 | 78.35 | 58.90 | 73.16 | 76.05 | 45.14 | 32.80 | 63.22 |
| | PQP(ours) | 79.05 | 81.60 | **62.30** | 73.01 | **78.95** | **49.49** | **36.60** | **65.57** |
| | Dense | 81.23 | 84.15 | 65.45 | 77.35 | 81.00 | 52.82 | 38.00 | 68.57 |
| 1-65B | SparseGPT | 80.03 | 84.40 | 63.25 | 76.64 | 79.15 | 49.23 | 36.40 | 67.01 |
| | Wanda | 79.82 | 82.95 | 62.90 | 75.93 | 79.30 | 49.91 | 35.80 | 66.66 |
| | RIA | 79.49 | 83.25 | 61.95 | 76.09 | 79.05 | 47.44 | 35.60 | 66.13 |
| | BESA | 80.30 | **84.45** | **65.10** | 76.09 | 79.45 | 50.43 | 37.60 | 67.63 |
| | BaWA | 79.76 | 83.50 | 62.10 | 75.77 | 79.20 | 48.38 | 35.20 | 66.27 |
| | PQP(ours) | **80.58** | 83.65 | 64.85 | **77.90** | **80.25** | **51.79** | **37.80** | **68.12** |

Table 14: Zero-shot accuracies (%) of post-quantization pruning LLaMA2 models on the PIQA, BoolQ, HellaSwag, WinoGrande, ARC, and OBQA datasets with unstructured 50% sparsity. 2-7/13/70B represents LLaMA2-7/13/70B models. The **best** performing result is indicated in **bold**, while the second best is underlined.

| Models | Methods | PIQA | BoolQ | HellaSwag | WinoGrande | ARC-e | ARC-c | OBQA | Average |
|---|---|---|---|---|---|---|---|---|---|
| | Dense | 78.07 | 77.03 | 57.13 | 69.06 | 76.30 | 43.43 | 31.40 | 61.77 |
| 2-7B | SparseGPT | 75.08 | 73.36 | 51.24 | **67.96** | 68.39 | 34.98 | 27.40 | 56.92 |
| | Wanda | 75.41 | 73.67 | 49.74 | 64.80 | 69.57 | 36.26 | 29.00 | 56.92 |
| | RIA | 74.43 | **74.68** | 48.97 | 66.30 | 67.80 | 33.36 | 28.00 | 56.22 |
| | BESA | 76.22 | 73.91 | 52.27 | 65.90 | 71.13 | **38.48** | 28.40 | 58.04 |
| | BaWA | 73.56 | 73.36 | 49.19 | 66.46 | 68.35 | 33.96 | 25.20 | 55.73 |
| | PQP(ours) | **76.28** | 74.56 | **53.12** | 66.61 | **71.76** | 38.05 | **29.40** | **58.54** |
| | Dense | 79.05 | 80.06 | 60.05 | 72.22 | 79.38 | 48.46 | 35.20 | 64.92 |
| 2-13B | SparseGPT | 76.71 | 79.14 | 55.31 | **72.14** | 73.86 | 39.25 | 30.20 | 58.09 |
| | Wanda | 76.71 | 77.37 | 55.24 | 70.48 | 74.87 | 41.98 | 32.20 | 61.26 |
| | RIA | 76.33 | **80.98** | 54.22 | 69.38 | 74.16 | 39.16 | 30.40 | 60.66 |
| | BESA | 77.37 | 78.41 | 57.29 | 70.72 | **75.72** | 42.66 | **32.80** | 62.14 |
| | BaWA | 76.66 | 80.49 | 54.40 | 70.32 | 74.83 | 40.19 | 30.20 | 61.01 |
| | PQP(ours) | **78.02** | 80.76 | **57.43** | 70.40 | 75.51 | **42.75** | 32.00 | **62.41** |
| | Dense | 82.15 | 83.15 | 66.05 | 77.98 | 82.55 | 54.35 | 37.20 | 69.06 |
| 2-70B | SparseGPT | 81.39 | 83.20 | 62.90 | **78.22** | **81.35** | 51.71 | 36.40 | 67.88 |
| | Wanda | 80.85 | 83.80 | 62.95 | 77.43 | 79.65 | 50.34 | 34.60 | 67.09 |
| | RIA | 80.85 | 84.30 | 62.25 | 76.40 | 80.30 | 50.26 | 34.60 | 66.99 |
| | BESA | 81.61 | **84.70** | 64.15 | 76.56 | 80.40 | **52.39** | 37.60 | 68.20 |
| | BaWA | 80.85 | 84.05 | 62.30 | 76.48 | 80.30 | 49.74 | 35.20 | 66.99 |
| | PQP(ours) | **81.77** | 83.65 | **64.90** | 77.35 | 80.50 | 51.96 | **38.20** | **68.33** |