# OpenReview forum: "PQP: A one-shot collaborative method for post-quantization pruning of LLMs"
_ICLR.cc/2026/Conference — ICLR 2026 Conference Withdrawn Submission_

### Official Review · Reviewer_6pDF · 2025-10-22

**Soundness:** 3
**Presentation:** 2
**Contribution:** 2
**Rating:** 2
**Confidence:** 4

**Summary:**

The authors propose a novel method, PQP, that performs both quantization and pruning in one-shot without any re-training. The core idea behind the method is to minimize the layer-wise reconstruction error of the quantized and pruned model with respect to the pruning mask and sparsity rates. The authors propose an objective to minimize as well as solutions to tackle the non-differentiability and challenges of convergence. The authors benchmark their method across a variety of downstream tasks against pruning methods applied to quantized models showing that PQP is able to outperform prior pruning methods.

**Strengths:**

1. The authors provide a unified framework to performing both quantization and pruning.
2. The performance benchmarks of PQP seem to be very competitive and outperform approaches where pruning is merely applied naively after quantization.
3. The method does not require re-training and can be done in one-shot — providing a low-cost alternative to methods that would require re-training.

**Weaknesses:**

1. No intuition/reasoning is provided behind the PQP metric.
2. Equation 13 needs more exposition and is currently unmotivated. Why is this a reasonable thing to be doing?
3. The algorithm has **a lot** of hyperparameters that are not fully ablated in the paper. For example, there are a set of hyperparameters associated with optimizing the objective Equation 9 that are not fully addressed. The choice of D and how the {r_d} should be chosen/spaced are also not fully addressed either. This is particularly important since the spacing of {r_d} seems to affect Equation 13 and it’s unclear how things would change if {r_d} is varied.
4. The method is only tested on dated models (LLaMA and LLaMA-2) and a single model family which potentially leads to questions as to whether the method generalizes to more recent models and different model families.
5. PQP is formulated in a way such that pruning is done post-quantization. This has been shown to be suboptimal in [1] where the authors show (theoretically and empirically) that pruning should be done prior to quantizing.
6. A lot of effort is put into the paper to simplify the objective (Equation 3) in order to get better convergence but there is nothing in the paper that suggests that the simplified objective (Equation 8) actually converges or if it is superior to the original formulation without the simplification.
7. In the supplementary material (Table 10), it seems to show that increasing the training epochs of Equation 9 can actually lead to worse perplexity performance. This is counter-intuitive and raises into question as to whether Equation 9 is actually being minimized practically.
8. The clarity of Figure 1 could be improved a it is currently unclear without reading the inline text.

**Questions:**

1. Could the authors provide some intuition/motivation as to why the PQP metric is a reasonable metric to be using for the pruning metric? Equation 13 also warrants additional motivation.
2. In the results for PQP, how many epochs were used to optimize the objective (Equation 9)?
3. Could the authors include a plot that shows the objective (Equation 9) decreasing with training epochs?
4. Did you ablate minimizing Equation 4 directly versus minimizing Equation 8?
5. In Tables 2/3/4, are the pruning metrics that PQP is being compared to just being computed directly using the quantized model? Since PQP is made aware of the quantization-step, this comparison is a bit unfair, as the other pruning algorithms are operating directly on the quantized model and unaware of the quantization that was performed. Are there any methods that the authors can compare to that combine both pruning and quantization in one single method? (If no such method exists, this should be emphasized in the paper).
6. There are various instances of where \citet and \citep are misused.
7. Is the normality assumption on B necessary? Don't you simply need the first moment to be 0? Could the authors provide empirical evidence to justify that the expectation of B is zero?
8. Wouldn't it make more sense to put the PQP Algorithm bubble in the main text as opposed to the appendix?
9. The authors might want to encapsulate Section 3.3 into a theorem box accompanied with the proof.
10. It might be better to clarify what W is in Table 1 (i.e. are the pruning scores being computed directly with the quantized models?)
11. Are there perhaps any more recent pruning methods that the authors are not comparing their method to?

[1] EFFECTIVE INTERPLAY BETWEEN SPARSITY AND QUANTIZATION: FROM THEORY TO PRACTICE

---

### Official Review · Reviewer_qc9r · 2025-10-26

**Soundness:** 3
**Presentation:** 3
**Contribution:** 3
**Rating:** 4
**Confidence:** 5

**Summary:**

The authors of this manuscript propose PQP (Post-Quantization Pruning), a one-shot, training-free method to collaboratively prune already-quantized LLMs. The core idea is to introduce a new pruning metric (PQP-Metric) specifically designed for quantized weights and to integrate the statistical properties of quantization error into a block-wise pruning loss. This allows the framework to remove weights that are either unimportant or heavily impacted by quantization noise, all without requiring fine-tuning.

**Strengths:**

1- The paper is well-written and the proposed PQP method is clearly explained and easy to follow.

2- The one-shot, training-free nature of the framework is a significant strength, making it practical and computationally efficient for real-world deployment.

3- The concept of a pruning metric (PQP-Metric) that is aware of the post-quantization state of the weights is a novel and well-motivated contribution.

4- The reported perplexity (Table 2) and zero-shot accuracy (Table 3) results show consistent improvements over baseline one-shot pruning methods (Wanda, SparseGPT, etc.) applied on top of quantized models.

**Weaknesses:**

1- The comparison results are missing a crucial benchmark, JSQ [1], which is one of the closest and most relevant works on joint one-shot sparsification and quantization. Without this comparison, it's difficult to evaluate the true state-of-the-art performance of PQP.

2- The models used for evaluation (LLaMA and LLaMA 2) are known to be relatively robust to compression. Based on my experience, a stronger evaluation would include more recent and sensitive models, such as the LLaMA 3.x or Gemma 3 families, to truly test the method's robustness.

3- The main results (Tables 2 and 3) focus on 50% unstructured sparsity, which is notoriously difficult to accelerate on GPUs. While 2:4 sparsity is briefly explored for the metric (Table 4), the paper lacks a comparison to state-of-the-art semi-structured sparsity methods like MaskLLM [2] or ProxSparse [3], which are designed for practical speedups.

4- A more detailed analysis of the overhead is needed. While Section 4.3 discusses the low parameter overhead of learning the pruning rates, it does not provide a clear picture of the time overhead of this block-wise pruning process.

---

[1] Guo et al., Compressing Large Language Models by Joint Sparsification and Quantization, ICML 2024

[2] Fang et al., MaskLLM: Learnable Semi-Structured Sparsity for Large Language Models, NeurIPS 2024

[3] Liu et al., PROXSPARSE: REGULARIZED LEARNING OF SEMI-STRUCTURED SPARSITY MASKS FOR PRETRAINED LLMS, ICML 2025

**Questions:**

1- How does PQP compare against JSQ [1] in a head-to-head comparison on the same models and compression settings?

2- For semi-structured 2:4 sparsity, how does PQP (using its row-wise metric) compare against specialized methods like MaskLLM [2] and ProxSparse [3]?

3- Is the PQP framework compatible with low-rank approximation (LoRA) methods? For example, could it be combined with a method like SLiM [4] for even greater compression or higher accuracy?

4- Given the focus on 50% unstructured sparsity, what are the practical, measured inference speedups (if any) on a GPU? Or is the primary benefit of PQP at this sparsity level limited to storage reduction?

---

[1] Guo et al., Compressing Large Language Models by Joint Sparsification and Quantization, ICML 2024

[2] Fang et al., MaskLLM: Learnable Semi-Structured Sparsity for Large Language Models, NeurIPS 2024

[3] Liu et al., PROXSPARSE: REGULARIZED LEARNING OF SEMI-STRUCTURED SPARSITY MASKS FOR PRETRAINED LLMS, ICML 2025

[4] Mozaffari et al., SLiM: One-shot Quantization and Sparsity with Low-rank Approximation for LLM Weight Compression, ICML 2025

**Details Of Ethics Concerns:**

No concerns.

---

### Official Review · Reviewer_Qx1R · 2025-10-27

**Soundness:** 2
**Presentation:** 3
**Contribution:** 3
**Rating:** 4
**Confidence:** 3

**Summary:**

The paper proposes PQP (Post-Quantization Pruning), a one-shot pruning pipeline that first performs post-training quantization (PTQ) and then prunes the already-quantized weights. The key ideas are: a PQP-Metric that scores importance for quantized weights by multiplying their magnitude with a pruning metric and a simplified block-level reconstruction loss with learnable differentiable sparsity. Experiments show consistent perplexity and zero-shot improvements over previous methods, compressing the model size to one eighth of its original size (4-bit × 50% sparsity) while maintaining performance. The metric can be also transferred to 2:4 and 4:8 semi-structured sparsity.

**Strengths:**

1. The article presents and defines a clear issue: it well clarifies the challenge of compound errors when naively combining PTQ and pruning, and addresses this problem by pruning metrics and explicitly considering quantified losses.
2. The PQP-Metric combines quantized weights and pruning metric from the original weights, which is a clean way to inject pre-quantization importance while operating on quantized parameters. It avoids the complex computational load of gradients/Hessian inverses, and can be easily combined with existing methods or pipelines.
3. Across LLaMA/LLaMA-2 families of different sizes, PQP consistently achieves the lowest perplexity at 50% sparsity after 4-bit quantization and improves zero-shot averages. Meanwhile, under semi-structured constraints, this method can still maintain leading performance.

**Weaknesses:**

1. Equation (3) is derived from Eq. (1) by substituting $\(\hat{W_l} - B_l)$ for $\hat{W_l}$, but since $B_l = \hat W_l - W_l$, this collapses to masking $W_l$ and is **not equivalent** to Eq. (1). It should instead be $ \arg\min_{M_l}||X_l W_l - X_l\big((W_l + B_l)\odot M_l\big)||_F^2$, which preserves the intended incorporation of quantization error. Building on this decomposition, the validity of the later analysis and proofs hinges on an objective that appears to be derived from a non-equivalent substitution.
2. While performance are strong, the paper does not report latency or throughput in real scenarios, which are critical for claiming deployment benefits, especially since unstructured sparsity and 4-bit kernels can be hardware-limited.
3. Baseline scope is skewed toward pruning-only metrics applied after GPTQ.  It is necessary to supplement the results based on different quantification methods (such as OPTQ, AWQ) to verify the robustness of PQP. Meanwhile, given the pitch is joint compression,  comparison to recent joint sparsification+quantization frameworks (For example, the JSQ[1] method mentioned in the article) would strengthen claims.


[1] Jinyang Guo, Jianyu Wu, Zining Wang, Jiaheng Liu, Ge Yang, Yifu Ding, Ruihao Gong, Haotong Qin, and Xianglong Liu. Compressing large language models by joint sparsification and quantization. In Forty-first International Conference on Machine Learning, 2024.

**Questions:**

1. Results only use 4-bit GPTQ. How does PQP behave with other PTQ schemes (e.g., AWQ, OWQ, activation quantization, per-channel vs per-group)?
2. For unstructured and 2:4/4:8 sparsity, what are the actual end-to-end throughput and latency gains?
3. How many steps/epochs and how much GPU time are needed to learn the parameters compared to other pruning methods?

---

### Official Review · Reviewer_xPjA · 2025-11-01

**Soundness:** 2
**Presentation:** 3
**Contribution:** 2
**Rating:** 2
**Confidence:** 4

**Summary:**

The paper proposes Post-Quantization Pruning (PQP), a Quantization + Pruning method that improves sparsification of quantized models. The authors propose a novel saliency score for pruning that accounts for quantization. The paper integrates the quantization reconstruction error into the loss function to learn the sparsity mask and shows improvements on perplexity and downstream benchmarks with Llama and Llama-2 as the base models.

**Strengths:**

* The paper shows improvements over relevant methods across perplexity and downstream benchmarks.
* The proposed PQP metric is able to correct for the noise induced by quantization during saliency calculation.

**Weaknesses:**

* The paper evaluates on Llama and Llama-2, both of which are far away from the frontier open-source models in quality. The authors should evaluate on more recent models such as Qwen2.5/Qwen3 or DeepSeek.

**Questions:**

* Equation 4 can be replaced with learning a sparsity mask on the original weights with the PQP metric. What is the intuition behind framing it this way? Also, how does it compare with learning a sparsity mask on the original weights with the PQP metric.
* Do you see a tradeoff between the precision and the sparsity? That is, for a target memory requirement, what is the most optimal precision and sparsity ratio.
* How does PQP metric compare to PW(Quantized W)?
* How does the algorithm here compare with using PW(Quantized W) following by minimizing Equation 4 with only quantized weights (and not with the original weights).
* Are both weights and activations quantized?

---

### Note · Authors · 2026-01-16

I have read and agree with the venue's withdrawal policy on behalf of myself and my co-authors.